# Pessimism for Offline Linear Contextual Bandits using $\ell_p$ Confidence Sets

**Gene Li**
Toyota Technological
Institute at Chicago
gene@ttic.edu

**Cong Ma**
Department of Statistics
University of Chicago
congm@uchicago.edu

**Nathan Srebro**
Toyota Technological
Institute at Chicago
nati@ttic.edu

## Abstract

We present a family $\{\widehat{\pi}_p\}_{p \geq 1}$ of pessimistic learning rules for offline learning of linear contextual bandits, relying on confidence sets with respect to different $\ell_p$ norms, where $\widehat{\pi}_2$ corresponds to Bellman-consistent pessimism (BCP), while $\widehat{\pi}_\infty$ is a novel generalization of lower confidence bound (LCB) to the linear setting. We show that the novel $\widehat{\pi}_\infty$ learning rule is, in a sense, adaptively optimal, as it achieves the minimax performance (up to log factors) against all $\ell_q$-constrained problems, and as such it strictly dominates all other predictors in the family, including $\widehat{\pi}_2$.

## 1  Introduction

Offline (or batch) reinforcement learning (RL) [17, 18] seeks to learn a good policy from fixed historical data without active interactions with the environment. This offline paradigm has been widely adopted in applications including dialog generation [10], autonomous driving [43], and robotic control [16], etc.

When the offline dataset has insufficient coverage over the state and action spaces, planning via nominal estimates of either the value function or the model may perform poorly—a phenomenon that is observed even in a simple two-armed bandit [24]. This challenge motivates the adoption of the *pessimism principle* for solving offline RL. In essence, the pessimism principle discounts policies that are less represented/supported in the offline dataset, and hence is pessimistic/conservative in outputting a policy. Built on this common principle, a diverse collection of pessimistic learning rules have been proposed in theory and practice [11, 24, 36, 37, 45, 46, 9, 15, 34, 14, 21, 42]. This leads us to the following natural question:

> *Which pessimistic learning rule should one use for solving offline RL problems?*

In this paper, we address the question in the setting of offline linear contextual bandits, in which the expected reward—as a function of the state-action pair—is linear with respect to a known feature mapping that maps state-action pairs to finite-dimensional vectors. Our goal is to make sense of previously proposed learning rules for offline RL, and understand which learning rule is "optimal" in a statistical sense. We present a general family $\{\widehat{\pi}_p\}_{p \geq 1}$ of pessimistic learning rules based on the construction of $\ell_p$ confidence sets for the unknown linear parameter. We advocate for $\widehat{\pi}_\infty$, a new $\ell_\infty$ learning rule for offline linear contextual bandits, which we call Pessimism via Uniform Norm Confidence (for short, PUNC).[1] PUNC directly extends the lower confidence bound algorithm proposed in the tabular contextual bandit setting [24]. We show that PUNC (1) achieves a suboptimality guarantee that dominates other $\widehat{\pi}_p$ (up to log factors, which we ignore throughout the introduction), and (2) has an *adaptive minimax optimality* property that is unique among the family $\{\widehat{\pi}_p\}_{p \geq 1}$. In particular, we argue that PUNC dominates prior learning rules which are based on $\ell_2$ pessimism (e.g., [37, 45, 11]) and which cannot attain adaptive minimax optimality.

---

[1] Throughout the paper, we use $\widehat{\pi}_\infty$ and PUNC interchangeably.

36th Conference on Neural Information Processing Systems (NeurIPS 2022).

**Roadmap.** We first introduce a broad class of pessimistic learning rules in Section 3. The construction of these pessimistic learning rules relies on the observation that any *confidence set* of the linear reward function automatically induces a pessimistic value estimate, and hence a pessimistic learning rule. As concrete examples, for each $p \geq 1$, one can design $\widehat{\pi}_p$, an $\ell_p$ learning rule, by constructing such a confidence set using the $\ell_p$ distance metric. We show in Section 3.3 that $\widehat{\pi}_2$ recovers the Bellman-consistent pessimism (BCP) learning rule [37], proposed for offline RL with general function approximation; meanwhile, $\widehat{\pi}_\infty$ generalizes the lower confidence bound (LCB) learning rule, proposed for offline tabular RL, to the linear setting.

Once we have cast pessimistic estimation in this framework, we can study the performance guarantees of the family $\{\widehat{\pi}_p\}_{p \geq 1}$. Employing a notion of *pessimism-validity* (Definition 1) allows us to easily to derive upper bounds on suboptimality for each $\widehat{\pi}_p$ in terms of the dual $\ell_q$ norm (where $1/p + 1/q = 1$); see Theorem 1. For $p = 2$, the upper bound improves over that provided in the paper [37] for linear contextual bandits. For $p = \infty$, the upper bound matches that proved in the paper [24] for tabular contextual bandits. A key observation regarding the upper bound is that the suboptimality guarantee of $\widehat{\pi}_\infty$ *dominates* all other $\widehat{\pi}_p$ in the general linear setting. This partially showcases the advantage of using PUNC.

To further investigate the advantage of PUNC over other $\widehat{\pi}_p$ (for $p \in [1, \infty)$), we consider the fundamental statistical limits of the offline linear contextual bandit problem in Section 4. Inspired by both the upper bounds we prove and prior work [45, 24, 40], we consider a sequence of norm-constrained classes of contextual bandit instances indexed by the $\ell_q$ norm ($q \geq 1$). We prove that each $\widehat{\pi}_p$ is minimax rate-optimal within the dual $\ell_q$-norm constrained contextual bandit class; see Theorem 2. However, Theorem 2 delivers an even stronger message: PUNC is *adaptively minimax optimal* in the sense that it simultaneously achieves optimality for all $\ell_q$-norm constrained classes, as illustrated by Figure 1. We also demonstrate that such adaptivity is unique to PUNC as other values of $p$ (e.g., $p = 2$) cannot achieve simultaneous optimality. Instead, $\widehat{\pi}_p$ is only adaptively optimal for $\ell_q$-norm constrained classes where $q \geq p/(p-1)$; see Theorem 3.

In summary, our main contributions are the following:

- We introduce a novel learning rule, PUNC, for solving the offline linear contextual bandit problem, whose performance guarantee dominates those of all other $\widehat{\pi}_p$, for finite $p$ (Theorem 1).

- We show minimax lower bounds over norm-constrained classes of contextual bandit instances, which show that each $\widehat{\pi}_p$ is optimal over the dual $\ell_q$ class, up to log factors in the dimension (Theorem 2).

- We demonstrate that PUNC satisfies the adaptive minimax optimality property (Section 4.2), and show that this property is unique to PUNC by proving a separation result against any other $\widehat{\pi}_p$ (Theorem 3, and see also Figure 1).

## 2 Problem setup

We begin by introducing the problem of offline learning in linear contextual bandits. Let $\mathcal{S}$ and $\mathcal{A}$ be the state space and the action space, respectively. Let $\phi : \mathcal{S} \times \mathcal{A} \to \mathbb{R}^d$ be a known feature mapping. In the offline setting, we observe a dataset $\mathcal{D} := \{(s_i, a_i, r_i)\}_{i=1}^n$, where the covariates $\{(s_i, a_i)\}_{i=1}^n$ are fixed and the rewards are drawn independently according to $r_i \sim R(s_i, a_i)$, where $R(s, a)$ is the reward distribution associated with the pair $(s, a)$. We assume that $R(s, a)$ is 1-subgaussian for every $(s, a)$ with mean reward $r(s, a) := \mathbb{E}[R(s, a)]$. Furthermore, we assume that the expected reward is linear in the sense that for every $(s, a)$ pair, $r(s, a) = \phi(s, a)^\top \theta^\star$ for some unknown parameter vector $\theta^\star \in \mathbb{R}^d$.

Let $\pi : \mathcal{S} \to \mathcal{A}$ be a deterministic policy. Fixing a (known) test distribution $\rho \in \Delta(\mathcal{S})$, we define the value of the policy $\pi$ (with respect to $\rho$) as

$$V(\pi) := \mathbb{E}_{s \sim \rho} \left[ r(s, \pi(s)) \right] = \mathbb{E}_{s \sim \rho} \left[ \phi(s, \pi(s))^\top \theta^\star \right]. \tag{1}$$

Correspondingly, we define the optimal policy $\pi^\star$ as

$$\pi^\star(s) := \arg\max_{a \in \mathcal{A}} r(s, a) = \arg\max_{a \in \mathcal{A}} \phi(s, a)^\top \theta^\star, \quad \text{for each } s \in \mathcal{S}. \tag{2}$$

The goal of offline learning in linear contextual bandits is to design a learning rule which takes as input a dataset $\mathcal{D}$ and outputs a policy $\widehat{\pi}$ that maximizes the value (1); in this paper we abuse notation and also denote the learning rule as $\widehat{\pi}$. We measure the suboptimality of $\widehat{\pi}$ using $V(\pi^\star) - V(\widehat{\pi})$.

# 3 Offline learning with pessimism

The pessimism principle has recently gained much attention in offline RL theory and practice. At a high level, pessimistic learning rules first construct a data-dependent estimate $\widehat{V}(\pi)$ of the true value function $V(\pi)$ that is pessimistic, i.e., $\widehat{V}(\pi) \leq V(\pi)$ for all $\pi$. Then, the learning rule proceeds to select the policy that maximizes this pessimistic value function, i.e.,

$$\widehat{\pi} := \arg\max_{\pi \in \Pi} \widehat{V}(\pi). \tag{3}$$

Here, $\Pi \subseteq \mathcal{A}^{\mathcal{S}}$ is some policy class that contains the optimal policy $\pi^\star$. To see why this choice of policy makes sense, let us decompose the suboptimality of $\widehat{\pi}$ as follows:

$$V(\pi^\star) - V(\widehat{\pi}) = \left(V(\pi^\star) - \widehat{V}(\pi^\star)\right) + \left(\widehat{V}(\pi^\star) - \widehat{V}(\widehat{\pi})\right) + \left(\widehat{V}(\widehat{\pi}) - V(\widehat{\pi})\right). \tag{4}$$

The middle term is non-positive by definition of $\widehat{\pi}$. Due to the pessimistic property of $\widehat{V}$, we also have $\widehat{V}(\widehat{\pi}) - V(\widehat{\pi}) \leq 0$, which yields the suboptimality upper bound

$$V(\pi^\star) - V(\widehat{\pi}) \leq V(\pi^\star) - \widehat{V}(\pi^\star). \tag{5}$$

Consequently, under the selection rule (3), a tight pessimistic value function $\widehat{V}$ induces a policy with small suboptimality.

## 3.1 Achieving pessimism by building confidence sets

As a general strategy, one can construct the pessimistic value estimator $\widehat{V}$ by building confidence sets for the linear parameter $\theta^\star$. Let $\Theta \subseteq \mathbb{R}^d$ be a *confidence set* that contains the true parameter $\theta^\star$. We can define the corresponding pessimistic value estimator

$$\widehat{V}(\pi) := \inf_{\theta \in \Theta} \mathbb{E}_{s \sim \rho}[\phi(s, \pi(s))^\top \theta], \tag{6}$$

and its associated policy learning rule $\widehat{\pi}_\Theta := \arg\max_{\pi \in \Pi} \widehat{V}(\pi)$. Here for simplicity we take $\Pi$ to be the class of all deterministic policies.

In essence, the confidence set $\Theta$ captures the amount of uncertainty we have about the ground truth $\theta^\star$. Once $\Theta$ is determined, we construct the value estimate $\widehat{V}(\pi)$ via the worst-case value of $\pi$ among all plausible linear parameters $\theta$ in the confidence set $\Theta$. It is immediate to see that under the assumption $\theta^\star \in \Theta$, one has $\widehat{V}(\pi) \leq V(\pi)$ for all $\pi$. In other words, the value estimator $\widehat{V}$ is indeed pessimistic. As a result, we can apply the general upper bound (5) to obtain

$$V(\pi^\star) - V(\widehat{\pi}_\Theta) \leq V(\pi^\star) - \widehat{V}(\pi^\star) = \sup_{\theta \in \Theta} \mathbb{E}_{s \sim \rho}[\phi(s, \pi^\star(s))^\top (\theta^\star - \theta)], \tag{7}$$

where the identity follows from the definition (6). Clearly, the "smaller" the confidence set, the smaller the bound on suboptimality. An extreme case is when $\Theta$ contains the singleton $\theta^\star$, which yields zero suboptimality. However, since only noisy rewards are observed, we cannot hope to construct such a good confidence set. Given the uncertainty about the rewards, our confidence set has to be "large" enough in order to guarantee that $\theta^\star \in \Theta$ with decent probability.

Below we present a general definition called pessimism-validity that involves both the size of the confidence set and also its confidence level, both of which allow us to bound the suboptimality of the pessimistic learning rule $\widehat{\pi}_\Theta$. Let $\|\cdot\|$ be any norm over $\mathbb{R}^d$ that will be used to measure the size of the confidence set $\Theta$. Let $\delta \in (0, 1)$ be the failure probability. We have the following definition.

**Definition 1.** *We say the confidence set $\Theta$ is $(\beta, \delta)$ pessimism-valid under the norm $\|\cdot\|$ if with probability at least $1 - \delta$, the following two properties hold: (1) $\theta^\star \in \Theta$; (2) $\sup_{\theta \in \Theta} \|\theta^\star - \theta\| \leq \beta$.*

A $(\beta, \delta)$ pessimism-valid confidence set $\Theta$ automatically induces a pessimistic learning rule $\widehat{\pi}_\Theta$ with bounded suboptimality, as shown in the following proposition.

**Proposition 1.** *Suppose that $\Theta$ is $(\beta, \delta)$ pessimism-valid under the norm $\|\cdot\|$. Then with probability at least $1 - \delta$, the pessimistic learning rule $\widehat{\pi}_\Theta$ obeys*

$$V(\pi^\star) - V(\widehat{\pi}_\Theta) \leq \beta \cdot \left\| \mathbb{E}_{s \sim \rho}[\phi(s, \pi^\star(s))] \right\|_*,$$

*where $\|\cdot\|_*$ is the dual norm of $\|\cdot\|$.*

Proposition 1 simply follows from the upper bound (7), the definition of pessimism-validity, and the definition of the dual norm.

## 3.2 Building $\ell_p$ confidence sets

In this section, we instantiate the general strategy introduced above for achieving pessimism by constructing an $\ell_p$ confidence set around the true parameter $\theta^\star$ for some $p \geq 1$. Such constructions using $\ell_p$ norms include the aforementioned BCP and LCB learning rules (as well as other recently proposed learning rules) as special cases. As we will see, setting up the notion of pessimism-validity allows us to easily bound the suboptimality of the induced policy learning rules.

Let us denote the data matrix $\Phi \in \mathbb{R}^{n \times d}$, where the $i$-th row of $\Phi$ is given by $\phi(s_i, a_i)^\top$. We also define the observed reward vector $r := (r_1, \ldots, r_n)^\top \in \mathbb{R}^n$. Let $\widehat{\theta}_{\text{ols}} := (\Phi^\top \Phi)^{-1} \Phi^\top r$ be the ordinary least-squares estimate for the true parameter $\theta^\star$. Throughout the paper, we assume that the sample "covariance" matrix $\Sigma_{\mathcal{D}} := \frac{1}{n} \sum_{i=1}^n \phi(s_i, a_i) \phi(s_i, a_i)^\top = \frac{1}{n} \Phi^\top \Phi$ is invertible. (The results in the paper can be modified to accomodate the scenario when $\Sigma_{\mathcal{D}}$ is not invertible by considering regularized quantities $\Sigma_{\mathcal{D}} + \lambda I$ for some $\lambda > 0$.) We then consider the confidence sets of the form:

$$\Theta_p := \left\{ \theta \in \mathbb{R}^d \mid \left\| \Sigma_{\mathcal{D}}^{1/2} (\theta - \widehat{\theta}_{\text{ols}}) \right\|_p \leq \beta/2 \right\}, \tag{8}$$

where $\beta > 0$ is a width parameter. In other words, the set $\Theta_p$ contains all the $\theta$'s that are close to the OLS estimate $\widehat{\theta}_{\text{ols}}$ in $\ell_p$ distance after the linear transformation $\Sigma_{\mathcal{D}}^{1/2}$. Since $\widehat{\theta}_{\text{ols}}$ is a faithful approximation of the truth $\theta^\star$, we expect that $\theta^\star$ lies in this confidence set $\Theta_p$ with an appropriate choice of $\beta$. This is indeed true, as the following lemma shows.

**Lemma 1.** *Fix any $\delta \in (0, 1)$. Set the width parameter $\beta = d^{1/p} \sqrt{\frac{8 \log(d/\delta)}{n}}$. Then the confidence set $\Theta_p$ is $(\beta, \delta)$ pessimism-valid with respect to the norm $\|v\| := \|\Sigma_{\mathcal{D}}^{1/2} v\|_p$.*

See Appendix B.1 for the proof of this lemma.

Combining Lemma 1 and Proposition 1 immediately yields the following performance guarantee for the pessimistic learning rule constructed using $\Theta_p$ (which for notational brevity we denote as $\widehat{\pi}_p$).

**Theorem 1.** *For any $p \geq 1$, with probability at least $1 - \delta$, we have*

$$V(\pi^\star) - V(\widehat{\pi}_p) \leq d^{1/p} \sqrt{\frac{8 \log(d/\delta)}{n}} \cdot \left\| \Sigma_{\mathcal{D}}^{-1/2} \mathbb{E}_{s \sim \rho} [\phi(s, \pi^\star(s))] \right\|_q,$$

*where $q$ is the solution to $1/p + 1/q = 1$.*

Several remarks regarding Theorem 1 are in order. First, the performance upper bound has a natural scaling w.r.t. the sample size $n$, i.e., $V(\pi^\star) - V(\widehat{\pi}_p) \lesssim \sqrt{1/n}$. In addition, Theorem 1 provides a family of upper bounds for each specific choice of $p \geq 1$. Lastly, from an upper bound perspective, the $\widehat{\pi}_\infty$ learning rule (which we call PUNC) dominates all the other $p \in [1, \infty)$, since for any $v \in \mathbb{R}^d$ and $q \in [1, \infty)$, the inequality $\|v\|_1 \leq d^{1-1/q} \|v\|_q$ holds. This partially showcases the benefits of using PUNC over the alternatives. Later in Section 4, we will see a stronger motivation—from the perspective of the lower bound—for using PUNC, in which we show that PUNC is adaptively minimax optimal. We also remark that the max-min form for $\widehat{\pi}_p$ has an equivalent max-only formulation, which will be helpful for our proofs and comparisons to other algorithms:

$$\widehat{\pi}_p = \arg\max_{\pi \in \Pi} \left\{ \mathbb{E}_{s \sim \rho} [\phi(s, \pi(s))]^\top \widehat{\theta}_{\text{ols}} - \frac{\beta}{2} \cdot \left\| \Sigma_{\mathcal{D}}^{-1/2} \mathbb{E}_{s \sim \rho} [\phi(s, \pi(s))] \right\|_q \right\}. \tag{9}$$

## 3.3 Connections to prior pessimistic learning rules

Now we present several connections to existing methods used for offline linear contextual bandits.

**Connection between $\widehat{\pi}_2$ and Bellman-consistent pessimism.** Xie et al. [37] proposed the idea of Bellman-consistent pessimism (BCP) for solving offline reinforcement learning with general function approximation. When specialized to linear contextual bandits, the BCP learning rule first forms a

version space that includes all possible linear reward functions with small $\ell_2$ prediction error on the observed datasets. Then, BCP defines each policy's pessimistic value as the smallest value the policy can achieve in the version space. Finally, BCP returns the policy that has the highest pessimistic value. In fact, BCP exactly matches the learning rule $\widehat{\pi}_2$ proposed herein. To see this, it suffices to note that the empirical estimate of the Bellman error (cf. Equation (3.1) in the paper [37]) in the linear contextual bandit case is given by

$$\frac{1}{n}\sum_{i=1}^n \big(\phi(s_i,a_i)^\top\theta - r_i\big)^2 - \inf_{\theta\in\mathbb{R}^d}\frac{1}{n}\sum_{i=1}^n\big(\phi(s_i,a_i)^\top\theta - r_i\big)^2 = \left\|\Sigma_{\mathcal{D}}^{1/2}(\theta - \widehat{\theta}_{\mathsf{ols}})\right\|_2^2.$$

Therefore a parameter $\theta$ having a small Bellman error is equivalent to having a small $\ell_2$ distance to the OLS estimate. Xie et al. [37] prove that BCP enjoys the guarantee (up to log factors) of $\sqrt{d/n}\cdot\mathbb{E}_{s\sim\rho}\big[\|\Sigma_{\mathcal{D}}^{-1/2}\phi(s,\pi^\star(s))\|_2\big]$, which is loose compared to our theoretical guarantee $\sqrt{d/n}\cdot\|\Sigma_{\mathcal{D}}^{-1/2}\mathbb{E}_{s\sim\rho}\phi(s,\pi^\star(s))\|_2$, as a consequence of Jensen's inequality and the convexity of the $\ell_2$ norm.

Similar ideas using the $\ell_2$ confidence set also appear in a recent paper by Zanette et al. [45]; their actor-critic algorithm, PACLE, can be interpreted as providing a computationally efficient way to solve $\widehat{\pi}_2$.[2]

**Connection between PUNC and lower confidence bound for tabular contextual bandits.** We now discuss how the LCB learning rule for the tabular setting is a specialization of PUNC. The tabular contextual bandit setting is a special case of the linear setting with the feature mapping $\phi(s,a) = e_{sa}$ (the canonical basis vector indexed by $(s,a)$). For notational convenience, we define $S := |\mathcal{S}|$, $A := |\mathcal{A}|$, $\widehat{r}(s,a)$ to be the empirical average reward, and $n(s,a)$ to be the number of times the pair $(s,a)$ is seen in the dataset.

Rashidinejad et al. [24] present the following lower confidence bound (LCB) learning rule:

$$\text{for each } s, \qquad \widehat{\pi}_{\mathsf{LCB}}(s) := \arg\max_{a\in\mathcal{A}} \widehat{r}(s,a) - \beta\cdot\sqrt{\frac{\log(SA/\delta)}{n(s,a)}}. \qquad (10)$$

In essence, the quantity $\widehat{r}(s,a) - \beta\cdot\sqrt{\frac{\log(SA/\delta)}{n(s,a)}}$ acts as a lower confidence bound for the true mean reward $r(s,a)$. In every state, LCB picks the action that maximizes this lower confidence bound. It is easy to verify that LCB (10) exactly corresponds to PUNC (with proper choices of $\beta$); one just needs to check the max-only formulation in Equation (9) with $p = \infty$ and $q = 1$.

In establishing performance guarantees for LCB, Rashidinejad et al. [24] assume that the covariates $\{(s_i,a_i)\}_{i=1}^n$ are drawn i.i.d. from a behavior distribution $\mu\in\Delta(\mathcal{S}\times\mathcal{A})$ (as opposed to our fixed design setting). Nevertheless, it is straightforward to translate our results to this random design case by using Chernoff bounds.

**Corollary 1.** *In the tabular setting, with probability at least $1-\delta$, the learning rule $\widehat{\pi}_\infty$ with $\Theta$ given by Equation (8) achieves the suboptimality:*

$$V(\pi^\star) - V(\widehat{\pi}_\infty) \lesssim \sqrt{\frac{\log(SA/\delta)}{n}}\cdot\left(\sum_s\frac{\rho(s)}{\sqrt{\mu(s,\pi^\star(s))}}\right),$$

*as long as $n\gtrsim\log(S/\delta)\cdot(\min_s\{\mu(s,\pi^\star(s))\})^{-1}$.*

Corollary 1 is proved in Appendix B.2.

Compared to the upper bound in the paper [24], Corollary 1 is more fine-grained, or "problem-dependent", as the suboptimality bound depends on the interaction between the specific behavior distribution $\mu$ and test distribution $\rho$. In contrast, Rashidinejad et al. [24] consider the class of tabular instances with bounded single-policy concentrability coefficient.

**Definition 2.** *The single-policy concentrability coefficient is defined as $C^\star := \sup_{s\in\mathcal{S}}\frac{\rho(s)}{\mu(s,\pi^\star(s))}$.*

Corollary 1 readily recovers the performance guarantee for LCB of $\tilde{O}(\sqrt{SC^\star/n})$ established in the paper [24], which is optimal in the regime where $C^\star \geq 2$.

---

[2]While we focus on the statistical properties of the family $\{\widehat{\pi}_p\}$ in this work, we believe that the actor-critic approach developed by Zanette et al. [45] can be extended to yield tractable algorithms for general $p \geq 1$.

**Connection to pessimistic value iteration.** We give another example of how to interpret pessimistic learning rules using the idea of confidence set construction. Consider the pessimistic value iteration (PEVI) learning rule proposed by Jin et al. [11]. PEVI directly extends Equation (10) to the linear setting:

$$\widehat{\pi}_{\mathsf{PEVI}}(s) := \arg\max_{a \in \mathcal{A}} \phi(s,a)^\top \widehat{\theta}_{\mathsf{ols}} - \beta \cdot \left\| \Sigma_{\mathcal{D}}^{-1/2} \phi(s,a) \right\|_2, \tag{11}$$

where the right hand side still acts as a lower confidence bound for the true mean reward $r(s,a)$. PEVI bears striking resemblance with the max-only formulation (9) (with $p = q = 2$), with the key difference that the max-only formulation is "averaged" over the test distribution $\rho$, while PEVI directly discounts every $(s,a)$ pair. PEVI does not immediately fit into our confidence set framework. However, if we modify the minimization over confidence sets to minimization over functionals $\theta : \mathcal{S} \to \mathbb{R}^d$, then we can rewrite PEVI as

$$\widehat{\pi}_{\mathsf{PEVI}} := \arg\max_{\pi \in \Pi} \inf_{\theta \in \Theta} \mathbb{E}_{s \sim \rho}[\phi(s, \pi(s))^\top \theta(s)],$$

$$\text{where } \Theta = \left\{ s \mapsto \theta(s) \mid \left\| \Sigma_{\mathcal{D}}^{1/2}(\theta(s) - \widehat{\theta}_{\mathsf{ols}}) \right\|_2 \le \beta, \text{ for all } s \right\}.$$

In other words, PEVI enlarges the $\ell_2$ confidence set by separately picking a pessimistic parameter $\theta(s)$ for each state $s \in \mathcal{S}$. Jin et al. [11] prove the guarantee (up to log factors) of $\sqrt{d^2/n} \cdot \mathbb{E}_{s \sim \rho} \left[ \| \Sigma_{\mathcal{D}}^{-1/2} \phi(s, \pi^\star(s)) \|_2 \right]$, which is loose due to the extra factor of $d$ and the interchanging of the expectation and the norm. However, their guarantee holds for all test distributions—as opposed to a fixed test distribution $\rho$. This is a consequence of being pessimistic for every state $s$.

# 4 Which learning rule should one use?

Having introduced a general strategy for building pessimistic learning rules by constructing $\ell_p$ confidence sets, it is natural to ask which $\widehat{\pi}_p$ one should use. To enable such comparisons, we investigate the statistical limits of offline linear contextual bandits over constrained sets of problem instances.

## 4.1 Minimax lower bound for constrained instances

For any feature mapping $\phi : \mathcal{S} \times \mathcal{A} \to \mathbb{R}^d$, sample size $n \in \mathbb{N}$, and two quantities $q \in [1, \infty)$, $\Lambda > 0$, we define a set of linear contextual bandit (CB) instances[3] as follows:

$$\mathsf{CB}_q(\Lambda) := \left\{ (\rho, \{(s_i, a_i)\}_{i=1}^n, \theta^\star, R) \mid \left\| \Sigma_{\mathcal{D}}^{-1/2} \mathbb{E}_{s \sim \rho}[\phi(s, \pi^\star(s))] \right\|_q \le \Lambda, \ R \text{ is 1-subgaussian} \right\}.$$

The set $\mathsf{CB}_q(\Lambda)$ includes all the linear contextual bandit instances such that a sort of "complexity measure" $\mathfrak{C}_q := \| \Sigma_{\mathcal{D}}^{-1/2} \mathbb{E}_{s \sim \rho}[\phi(s, \pi^\star(s))] \|_q$ is at most $\Lambda$. Our motivation to consider the rate of estimation in the CB family $\mathsf{CB}_q(\Lambda)$ are two-fold. First, in view of Theorem 1, the family $\mathsf{CB}_q(\Lambda)$ admits a good learning rule, specifically $\widehat{\pi}_p$ with $1/p + 1/q = 1$, since for every $\mathcal{Q} \in \mathsf{CB}_q(\Lambda)$, w.p. at least $1 - \delta$,

$$V_{\mathcal{Q}}^\star - V_{\mathcal{Q}}(\widehat{\pi}_p) \lesssim d^{1/p} \sqrt{\log(d/\delta)/n} \cdot \Lambda, \tag{12}$$

where $V_{\mathcal{Q}}^\star$ denotes the optimal value in instance $\mathcal{Q}$ and $V_{\mathcal{Q}}(\pi)$ denotes the value of policy $\pi$ in instance $\mathcal{Q}$. Thus, it is natural to view $\mathfrak{C}_q$ as a certain complexity measure for offline learning in linear contextual bandits. Second, prior work [45, 24, 40] has proven various types of lower bounds on offline learning using either the $\ell_2$ quantity $\mathfrak{C}_2$ or the $\ell_1$ quantity $\mathfrak{C}_1$. We will elaborate more on this point later.

Now we are ready to present the minimax lower bounds for these families of CB instances.

**Theorem 2.** *For every $d \ge 2$, there exists a feature mapping $\phi$ such that the following lower bound holds. For any $p, q \ge 1$ such that $1/p + 1/q = 1$, as long as $\Lambda \ge \sqrt{8} \cdot d^{1/q - 1/2}$ and $n \ge d^{2/p}\Lambda^2$, we have*

$$\inf_{\widehat{\pi}} \sup_{\mathcal{Q} \in \mathsf{CB}_q(\Lambda)} \mathbb{E}[V_{\mathcal{Q}}^\star - V_{\mathcal{Q}}(\widehat{\pi})] \ge c \cdot d^{1/p} \sqrt{1/n} \cdot \Lambda,$$

---

[3]For brevity, we omit the dependence on $\phi$ and $n$ in the notation $\mathsf{CB}_q(\Lambda)$.

*where $c > 0$ is some universal constant. Furthermore, when $p = \infty, q = 1$, the lower bound holds for the extended range of $\Lambda \geq 2$.*

The proof can be found in Appendix C. It relies on a reduction to a bound for the minimax regret of the multi-armed bandit problem.

We note that Theorem 2 also consists of a family of lower bounds for each $\ell_q$ norm constrained CB class. By comparing the lower bound in Theorem 2 with the upper bound (12) obtained by $\widehat{\pi}_p$, we see that for the $\ell_q$ norm constrained class $\mathsf{CB}_q(\Lambda)$, the learning rule $\widehat{\pi}_p$ with $1/p + 1/q = 1$ is minimax rate-optimal, up to a $\log d$ factor. For instance, over the $\ell_2$ class $\mathsf{CB}_2(\Lambda)$, the minimax rate of estimation is $\tilde{\Theta}(\sqrt{d/n} \cdot \Lambda)$, while over the $\ell_1$ class $\mathsf{CB}_1(\Lambda)$, the rate is given by $\tilde{\Theta}(\sqrt{1/n} \cdot \Lambda)$.

On a technical front, it would be interesting to extend Theorem 2 to the entire range of $\Lambda \geq 0$. It is unclear whether the same minimax rate of $\Omega(d^{1/p}/\sqrt{n} \cdot \Lambda)$ holds when $\Lambda = O(d^{1/q-1/2})$, or whether we can achieve faster rates in the small $\Lambda$ regime. In the tabular setting, Rashidinejad et al. [24] recently showed that LCB achieves fast $1/n$ rates when the single policy concentrability coefficient is small; similar results might hold in the linear setting. Several limitations prevent us from extending the range of $\Lambda$ in Theorem 2; Appendix C.1 provides more technical details.

## 4.2 Adaptive minimax optimality of PUNC

We point out a even stronger message delivered in Theorem 2: PUNC is *adaptively minimax optimal* for solving the offline linear contextual bandit problem. This is illustrated in Figure 1, where we plot the sample complexity $n$ required in order to achieve constant suboptimality (say, 0.01) for various $\mathsf{CB}_{p/(p-1)}(\Lambda)$. (For sake of illustration, it is more convenient to work with $p$ rather than $q$ on the $x$-axis.)

As indicated by the red line, Theorem 2 shows that every learning rule must incur sample complexity at least $\Omega(d^{2/p}\Lambda^2)$. Likewise, we can also follow the discussion after Theorem 1 to see that the performance upper bound of $\widehat{\pi}_\infty$ is $d^{1/p}\sqrt{\frac{\log(d/\delta)}{n}} \cdot \mathfrak{C}_q$ for all $p, q \geq 1, 1/p + 1/q = 1$. Thus, PUNC attains the green line in Figure 1; that is, PUNC is *simultaneously* minimax rate-optimal for all $\ell_q$-norm constrained classes $\mathsf{CB}_q(\Lambda)$, up to a $\log d$ factor.[4] From worst-case perspective, one should always prefer using PUNC given an unknown CB instance.

Is this adaptive optimality property unique to PUNC among the family $\{\widehat{\pi}_p\}_{p\geq 1}$ we consider? Below, we answer this question in the positive by presenting a separation result.

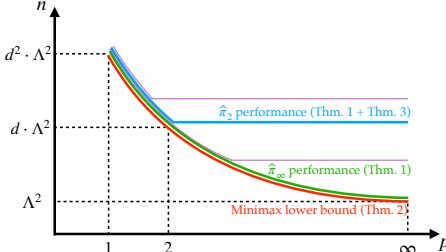

Figure 1: Sample complexity of $\widehat{\pi}_{\widetilde{p}}$ (for various $\widetilde{p}$) over different $\mathsf{CB}_{p/(p-1)}(\Lambda)$ classes. The red line corresponds the minimax lower bound. Other lines correspond to different values of $\widetilde{p}$ and show the number of samples $n$ required to ensure $\sup_{\mathcal{Q}\in\mathsf{CB}_{p/(p-1)}(\Lambda)} \mathbb{E}[V_{\mathcal{Q}}^\star - V_{\mathcal{Q}}(\widehat{\pi})] \leq 0.01$. The blue and green lines correspond to $\widehat{\pi}_2$ and $\widehat{\pi}_\infty$ respectively. Two purple lines correspond to $\widehat{\pi}_{\widetilde{p}}$ for some $\widetilde{p} \in (1,2)$ and $\widetilde{p} \in (2,\infty)$. PUNC attains minimax optimality over every class, while other $\widehat{\pi}_{\widetilde{p}}$ do not.

**Theorem 3** (Informal). *Fix any $p \geq 1$. For sufficiently large $n, d$, there exists a contextual bandit instance $\mathcal{Q} \in \mathsf{CB}_1(\Lambda)$ with $\Lambda = \sqrt{8d}$, such that with probability at least $1/4$, $\widehat{\pi}_p$ has suboptimality at least $\Omega(d^{1/p}/\sqrt{n} \cdot \Lambda)$.*

Since PUNC attains a suboptimality of $\tilde{O}(1/\sqrt{n} \cdot \Lambda)$ over the class $\mathsf{CB}_1(\Lambda)$, Theorem 3 shows that every other $\widehat{\pi}_p$ is *suboptimal* over the class $\mathsf{CB}_1(\Lambda)$.

A formal statement of Theorem 3 and its proof can be found in Appendix D. The key intuition in the proof is that the $\ell_p$ confidence sets capture a notion of error that is "averaged" over all directions, while the $\ell_\infty$ confidence sets separately estimate the error in each direction. In the hard instance we construct, only one direction determines the difficulty of the offline learning problem, so $\widehat{\pi}_p$ is worse

---

[4]We did not investigate when the $\log d$ factor in Theorem 1 can be removed, so for example, it is possible that $\widehat{\pi}_2$ beats $\widehat{\pi}_\infty$ over $\mathsf{CB}_2(\Lambda)$ by a factor of $\sqrt{\log d}$.

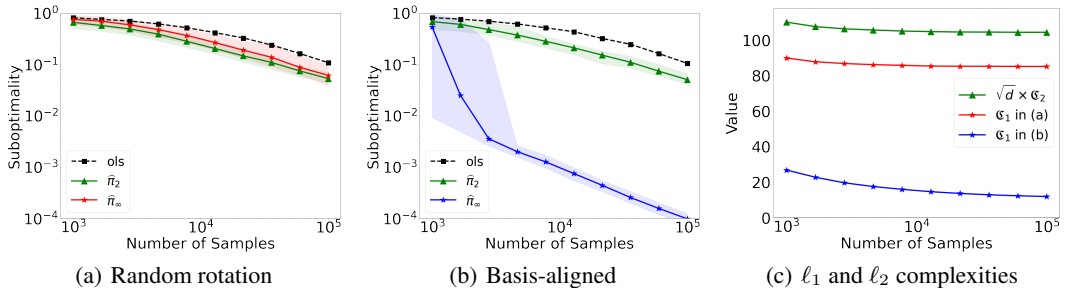

|   |   |   |
|---|---|---|
| (a) Random rotation | (b) Basis-aligned | (c) $\ell_1$ and $\ell_2$ complexities |

Figure 2: Comparing the performance of the plug-in rule, $\widehat{\pi}_2$, and $\widehat{\pi}_\infty$ on linear contextual bandit instances with $d = 100$, averaged over 100 trials, with 90% confidence intervals. (a) $\phi_i \sim \mathcal{N}(0, QDQ^\top)$ and $\theta^\star = Qe_{20}$, where $Q$ is a random rotation matrix and $D$ is a diagonal matrix with entries $D_{ii} = i^{-1}/(\sum_i i^{-1})$. (b) $\phi_i \sim \mathcal{N}(0, D)$ and $\theta^\star = e_{20}$. (c) computed average values for $\mathfrak{C}_1$ and $\sqrt{d} \times \mathfrak{C}_2$. The quantity $\mathfrak{C}_2$ is identical in both plots (a) and (b). For (a), $\mathfrak{C}_1 \approx \sqrt{d} \times \mathfrak{C}_2$, while for (b), $\mathfrak{C}_1 \ll \sqrt{d} \times \mathfrak{C}_2$.

by a factor of $d^{1/p}$. There is nothing special about the choice $\Lambda = \sqrt{8d}$, and our construction works for any $\Lambda \geq \Omega(\sqrt{d})$; we pick it to enable comparison with Theorem 2.

For sake of discussion, consider $\widehat{\pi}_2$. Theorem 1 shows that $\widehat{\pi}_2$ attains the rate:

$$
V(\pi^\star) - V(\widehat{\pi}_2) \lesssim
\begin{cases}
d^{1/p} \sqrt{\frac{\log(d/\delta)}{n}} \cdot \left\| \Sigma_{\mathcal{D}}^{-1/2} \mathbb{E}_{s\sim\rho}\left[\phi(s, \pi^\star(s))\right] \right\|_q & \text{when } q \geq 2, \\
\sqrt{\frac{d\log(d/\delta)}{n}} \cdot \left\| \Sigma_{\mathcal{D}}^{-1/2} \mathbb{E}_{s\sim\rho}\left[\phi(s, \pi^\star(s))\right] \right\|_q & \text{when } q \in [1, 2].
\end{cases}
$$

Together, Theorem 2 and 3 provide the message that both cases in the upper bound are tight (up to log factors). In the range $p \in [1, 2]$ (or $q \geq 2$), Theorem 2 shows that $\widehat{\pi}_2$ attains the minimax optimal rate (up to log factors) over $\mathsf{CB}_{p/(p-1)}(\Lambda)$, i.e., it is adaptively minimax optimal here. This explains the curved part of the blue line in Figure 1. On the other hand, Theorem 3 shows that $\widehat{\pi}_2$ cannot obtain the minimax rate over $\mathsf{CB}_1(\Lambda)$. Instead, $\widehat{\pi}_2$ may require $\Omega(d \cdot \Lambda^2)$ samples in order to achieve constant suboptimality. Since for any $p$, the set $\mathsf{CB}_{p/(p-1)}(\Lambda) \supseteq \mathsf{CB}_1(\Lambda)$, we know that $\widehat{\pi}_2$ may require $\Omega(d \cdot \Lambda^2)$ samples for any $\mathsf{CB}_{p/(p-1)}(\Lambda)$. Thus, the second case is tight when $p \geq 2$ (or $q \in [1, 2]$), explaining the flat part of the blue line in Figure 1. In general, for any finite $\widetilde{p}$, the learning rule $\widehat{\pi}_{\widetilde{p}}$ will be adaptively optimal for $\mathsf{CB}_{p/(p-1)}(\Lambda)$ only in the range $p \in [1, \widetilde{p}]$, and afterwards the sample complexity will "flatten out", as illustrated by the purple lines in Figure 1.

**Experimental Evidence.** In order to further validate this claim, we provide experimental evidence which shows that $\widehat{\pi}_2$ does not adapt to "easy" CB instances. In Figure 2, we consider a simple offline linear contextual bandit in which there is a single state and the feature set is $B_2^d$; thus the policy learning problem is equivalent to finding a vector $\pi \in \mathbb{S}^{d-1}$ that maximizes $V(\pi) := \pi^\top \theta^\star$. We vary the offline dataset distribution and the hidden parameter $\theta^\star$. When $\theta^\star$ is basis-aligned, we have $\mathfrak{C}_1 \ll \sqrt{d} \times \mathfrak{C}_2$; when $\theta^\star$ is not basis-aligned, the two quantities are on the same order.

### 4.3 Comparisons with prior lower bounds

There exist several lower bound results for offline reinforcement learning in the literature. In this section, we compare our lower bounds (cf. Theorem 2) with the prior bounds and highlight several improvements offered by our results.

**Comparison to lower bounds w.r.t. a single $\Lambda$.** Our lower bounds are stronger than those provided in the papers [11, 45], which hold for specific choices of $p = q = 2$ and a single fixed $\Lambda$. Take Theorem 2 of Zanette et al. [45] for example. Zanette et al. proved that the minimax rate of estimation over $\mathsf{CB}_2(\Lambda = d)$ is given by $d^{3/2}/\sqrt{n}$. Such a lower bound fails to uncover the fundamental scaling on the complexity $\Lambda$.[5] Theorem 4.7 of Jin et al. [11] shows a result in similar spirit; their construction essentially shows a minimax lower bound of $1/\sqrt{n}$ over $\mathsf{CB}_2(\Lambda)$ when $\Lambda = \Theta(1)$. Furthermore, their

---

[5] For instance, their result does not preclude the possibility that the correct lower bound over $\mathsf{CB}_2(\Lambda)$ takes an expression, say, $d^{-98.5}\Lambda^{100}/\sqrt{n}$.

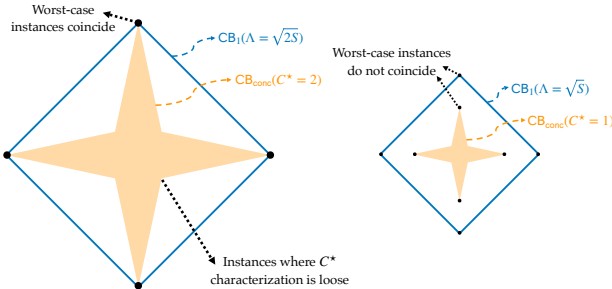

Figure 3: Illustrating the relationship between single policy concentrability and boundedness of $\mathfrak{C}_1$. Left: When $C^\star = 2$, the quantity $\mathfrak{C}_1$ always provides a tighter characterization of the problem difficulty, and the worst-case instances coincide. R: When $C^\star = 1$, the quantity $\mathfrak{C}_1$ does not provide a tight characterization in general.

lower bound is loose by a factor of $\sqrt{d}$ since they reduce to a two-point hypothesis testing problem. In contrast, our lower bound holds for nested families of CB instances with *varying* complexities $\Lambda$, which better showcases that the norm quantity is an intrinsic measure of difficulty for offline learning.

**Connections with single-policy concentrability.**   Our lower bound shares a similar flavor as that established in the paper [24], with the key difference lying in the class of CB instances under consideration: Rashidinejad et al. [24] consider the contextual bandit instances $\mathsf{CB}_{\mathrm{conc}}(C^\star)$ with bounded single-policy concentrability coefficient $C^\star$ (cf. Definition 2), while we consider the instances with bounded complexity $\mathfrak{C}_1$. These two quantities are intimately related, and we illustrate the relationship in Figure 3. As we have alluded to in Section 3.3, one has the inclusion

$$\mathsf{CB}_{\mathrm{conc}}(C^\star) \subseteq \mathsf{CB}_1(\sqrt{SC^\star}).$$

When $C^\star \geq 2$, the minimax rate of estimation over $\mathsf{CB}_{\mathrm{conc}}(C^\star)$ exactly matches that over $\mathsf{CB}_1(\sqrt{SC^\star})$, which implies that the hard instances for $\mathsf{CB}_{\mathrm{conc}}(C^\star)$ are also the hard instances in $\mathsf{CB}_1(\sqrt{SC^\star})$. However, this no longer holds when $C^\star \in [1, 2)$. Take $C^\star = 1$ as an example. Rashidinejad et al. show that the optimal rate over $\mathsf{CB}_{\mathrm{conc}}(C^\star = 1)$ is $S/n$, while Theorem 2 indicates that the optimal rate over $\mathsf{CB}_1(\Lambda = \sqrt{S})$ is $\sqrt{S/n}$. There is no contradiction, since the hard instances we construct for $\mathsf{CB}_1(\sqrt{S})$ satisfy $C^\star \geq 2$. This shows that when $C^\star < 2$, we "lose something" by working with the larger $\mathsf{CB}_1(\sqrt{SC^\star})$ class, as we are no longer able to achieve the fast rates possible over $\mathsf{CB}_{\mathrm{conc}}(C^\star)$.

On the flip side, the quantity $\mathfrak{C}_1$ can give tighter suboptimality guarantees than the $C^\star$ bound for a given instance. Consider the tabular instance where $\rho = \mathrm{Unif}(\mathcal{S})$ and $\mu(1, \pi^\star(1)) = 1/S^3$, while $\mu(s, \pi^\star(s)) = 1/S$ for all $s \geq 2$. This instance has $C^\star = S^2$, implying a guarantee of $S^{3/2}/\sqrt{n}$, while $\mathfrak{C}_1 = O(\sqrt{S})$, implying a better guarantee of $\sqrt{S/n}$.

## 4.4   A better complexity measure?

Our results lend support to the claim that we should always use PUNC, since it is simultaneously minimax rate-optimal over all the $\ell_q$ norm-constrained contextual bandit classes. Furthermore, the $\ell_1$ quantity $\mathfrak{C}_1$ can be thought of as a "complexity measure" that dominates other $\ell_q$ "complexity measures" $\mathfrak{C}_q$ for $q > 1$. To see this, consider the following thought experiment. Suppose before solving the linear contextual bandit problem, an oracle told us that the instance satisfies $\mathfrak{C}_q \leq \Lambda$. The results herein show that we do not lose anything by assuming that the instance satisfies the weaker condition $\mathfrak{C}_1 \leq d^{1/p}\Lambda$; using PUNC will give us the optimal rate of $d^{1/p}/\sqrt{n} \cdot \Lambda$.

However this is certainly not the complete answer to guiding question of "which pessimistic learning rule should one use for offline linear contextual bandits?". One piece of evidence comes from the comparison with the single policy concentrability assumption: in the regime where $C^\star \in [1, 2)$, we do "lose something" when we assume the instance satisfies the weaker condition $\mathfrak{C}_1 \leq \sqrt{SC^\star}$. Below we discuss another drawback associated with using $\mathfrak{C}_1$ as the complexity measure.

**Rotation ambiguity.**   One drawback of the complexity $\mathfrak{C}_1$ (as well as the learning rule PUNC) lies in the fact that it is not rotation invariant. (In fact, $\mathfrak{C}_2$ is the only rotational invariant complexity!)

To see this, let $U \in \mathbb{R}^{d \times d}$ be a fixed rotation matrix. Suppose that the features are rotated from $\phi$ to $U\phi$, which yields a different $\ell_1$ complexity $\mathfrak{C}_1(U) := \|U\Sigma_{\mathcal{D}}^{-1/2}\mathbb{E}_{s \sim \rho}[\phi(s, \pi^\star(s))]\|_1$, where $\Sigma_{\mathcal{D}}$ is defined using the old feature mapping. Since the $\ell_1$ norm is not rotation invariant, the $\mathfrak{C}_1(U)$ varies for differing choices of $U$, by up to a $\sqrt{d}$ factor. Thus, we cannot claim that any "complexity measure" $\mathfrak{C}_1(U)$ dominates others. A naive attempt to modify the $\ell_1$ set to be rotationally invariant by taking a minimization over $U$ also fails; observe that:

$$\Theta_1^{\min} := \left\{ \theta \in \mathbb{R}^d \mid \inf_U \left\| U\Sigma_{\mathcal{D}}^{1/2}(\theta - \widehat{\theta}_{\mathsf{ols}}) \right\|_1 \leq \beta \right\} = \left\{ \theta \in \mathbb{R}^d \mid \left\| \Sigma_{\mathcal{D}}^{1/2}(\theta - \widehat{\theta}_{\mathsf{ols}}) \right\|_2 \leq \beta \right\} =: \Theta_2,$$

that is, we recover $\widehat{\pi}_2$. A similar equivalence holds if we take the $\max$ inside the confidence set; we will recover the confidence set with an extra factor of $\sqrt{d}$.

**Instance-dependent optimality?**   Arguably, the strongest possible support for PUNC would be an instance-dependent lower bound which shows that for *every specific* linear contextual bandit instance, the performance achieved by PUNC is not improvable. Instance-dependent optimality results have been shown for related problems such as policy evaluation [23, 12] and optimal value estimation [13] in tabular MDPs; the recent work [7] also employs the local minimax method for online bandit and RL problems. For offline bandits, the paper [36] shows how a particular definition of instance optimality cannot be achieved by any algorithm. Establishing instance-dependent guarantees for offline learning is an important direction for future research.

Recent work [40] establishes the local minimax rate for offline learning in terms of the complexity $\mathfrak{C}_1$ for tabular contextual bandits. However, the theorem seems incorrect; we provide a counterexample in Appendix F to demonstrate—via explicit construction—that the complexity $\mathfrak{C}_1$ cannot characterize the local minimax risk for a two-armed bandit instance. The key observation is that the reduction used in the proof of the paper [40] from offline policy learning to optimal value estimation is invalid; if the gap in rewards for different actions is large, offline policy learning is fundamentally easier than optimal value estimation. This in turn allows us to break the claimed parametric $1/\sqrt{n}$ rate.

## 5   Conclusion

In this paper, we introduce a family $\{\widehat{\pi}_p\}_{p \geq 1}$ of pessimistic learning rules that include a number of prior works as special cases for the problem of offline learning in linear contextual bandits. We prove upper bounds for each learning rule $\widehat{\pi}_p$ and show matching minimax lower bounds over appropriately defined constrained instance classes. Our results highlight the benefits of using PUNC, the $\widehat{\pi}_\infty$ learning rule: namely (1) the guarantee for PUNC dominates all others; (2) PUNC is the sole learning rule with an adaptive minimax property. In particular, our results demonstrate that prior learning rules based on $\ell_2$ pessimism can be suboptimal (by a factor of $\sqrt{d}$).

Below we list several interesting directions for future investigation.

- *Extending to MDPs.* The MDP setting is more difficult due to the long horizon and transition dynamics. Extending the results of this paper to the MDP setting is an interesting future direction. One possible approach is to modify the PACLE algorithm [45] to solve for any $\ell_p$ learning rule.

- *Gap-dependent bounds.* In online RL, there is a wealth of results which adapt to easy instances which are characterized by gap structure in the rewards, see, e.g., [6, 32]. Obtaining tight gap-dependent bounds for the offline setting is an interesting direction for future work.

- *Offline RL with general function approximation.* In this paper, we focus on offline RL with linear function approximation. What is the right extension of these $\ell_p$ learning rules to general function approximation? While $\widehat{\pi}_2$ has the natural interpretation of defining a version space with small squared prediction error, no such interpretation exists for PUNC. It would be interesting to establish an analog for PUNC for general function classes.

## Acknowledgments and Disclosure of Funding

This work is supported by funding from the Institute for Data, Econometrics, Algorithms, and Learning (IDEAL).

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
