# OpenReview forum: "Pessimism for Offline Linear Contextual Bandits using $\ell_p$ Confidence Sets"
_NeurIPS.cc/2022/Conference — NeurIPS 2022 Accept_

### Official Review · Reviewer_9CB2 · 2022-07-10

**Rating:** 6
**Confidence:** 3
**Soundness:** 3 good
**Presentation:** 3 good
**Contribution:** 2 fair

**Summary:**

The paper studies batch (offline) learning in linear contextual bandits (with a realizability assumption that the expected reward of a context and an action is a linear function of their feature mapping given to the learner).

They analyze a class of pessimistic learning rules indexed by different l_p norms. For each learning rule, the learning rule is to (1) build a confidence set of the true parameter under l_p norm; (2) pessimistically select the policy that maximizes a lower confidence bound on the expected reward constructed using the confidence set.

They provide near-optimality analysis of the learning rules and show that the learning rule with l_{\infty} norm enjoys the best performance guarantee under their analysis.

The show that each learning rule under a norm is minimax optimal under the specific norm, within a norm-constrained class of contextual bandit instances (instances that are "easy" to learn).

They show that the learning rule under l_{\infty} nrom is minimax optimal under all norms, while the other learning rules are not.




**Questions:**

See weaknesses.

**Limitations:**

I did not find limitations and potential negative societal impact.

**Strengths And Weaknesses:**

Strengths:

The paper is well-written and easy to follow.

They provide instance-dependent and minimax optimality analysis of learning rules under different norms.

The discussion about the connections to prior works is interesting.

Weaknesses:

1. The proposed learning rules seem to have already been discussed in many existing offline RL literature, as also mentioned by the authors. The methodological contribution seems limited.

2. It would be great if the authors discuss and compare with prior works on offline learning in contextual bandit literature (e.g. [1,2,3] and their follow-ups).

3. The paper lacks empirical evaluation. And I doubt its real-world applicability. In particular, the theoretical analysis of the proposed learning rule relies heavily on a realizability assumption, which might not hold in most of the real-world problems. Also, the minimax optimality analysis is within some benign class of instances, which might not hold in real-world applications. It would be great if the paper shows some empirical evidence that the proposed learning is better compared with other rules and existing works [1,2,3].


Some minor issues:
why in eq (6) we use \min (instead of \inf) and in eq (7) we use \sup?

[1] Dudík, Miroslav, John Langford, and Lihong Li. "Doubly robust policy evaluation and learning." ICML (2011)
[2] Bottou, Léon, et al. "Counterfactual Reasoning and Learning Systems: The Example of Computational Advertising." Journal of Machine Learning Research 14.11 (2013).
[3] Swaminathan, Adith, and Thorsten Joachims. "Counterfactual risk minimization: Learning from logged bandit feedback." International Conference on Machine Learning. PMLR, 2015.

---

> ### Author Response · Authors · 2022-08-01
> **Thank you for your review.**
>
> We thank the reviewer for their comments and time.
>
> **Important corrections**:  The main proposed learning rule has *not* been previously discussed in the literature.  The main methodological contribution lies in the proposal of the $\ell_\infty$ learning rule in the general linear contextual bandit setting.  This has not been previously proposed.  The previously proposed rules for linear contextual bandits are BCP [4], PACLE [5], and PEVI [6], which we show are suboptimal and dominated by the proposed $\ell_\infty$ rule.
>
> Indeed, once we introduce the general framework for pessimism via confidence sets (which is a significant part of the novel contribution and what enabled the methodological advance), the relationship to prior work (LCB, BCP, PACLE, PEVI) becomes apparent, and one can see that the previously proposed LCB rule is a special case of our proposed novel rule.  But LCB is only for the tabular setting, not for the general linear setting.  And without our framework, seeing the more general rule in the more general linear contextual bandit setting is perhaps not so obvious (e.g., it wasn’t obvious to researchers who previously worked on offline linear contextual bandits/RL and suggested the suboptimal BCP, PACLE, and PEVI rules). By way of imperfect analogy, this is a bit like saying Exponentiated Gradient is not new because the Winnow algorithm (developed for learning a structureless finite class) can be seen as a special case of EG (when each hypothesis is encoded as an indicator, with a feature representation that encodes the hypothesis class).
>
> **Further responses:**
> 1. **Comparison to prior work on offline contextual bandits.** We thank the reviewer for bringing to our attention these additional related works on offline contextual bandits, we will include them in our revision. The papers [1] and [3] study approaches which use importance reweighting to estimate the values of policies from offline data. An inherent feature of importance weighting methods is that the behavior policy which generates the offline data is either known or approximated. In contrast, our method does not require knowledge of the behavior policy, and instead relies on the principle of pessimism to do offline learning. Another salient difference between our work and the papers [1-3] is the study of statistical optimality of the proposed methods. Our work confirms that our $\ell_\infty$ learning rule is in one sense statistically optimal  for offline linear contextual bandits, while these papers do not discuss the optimality of their methods for policy learning.
> 2. **Using inf vs min.** You are correct, we should use inf, since in general, the set $\Theta$ can be infinite.
>
> [4]: Xie, Cheng, Jiang, Mineiro, Agarwal. “Bellman-consistent Pessimism for Offline Reinforcement Learning.”
> [5]: Zanette, Wainwright, Brunskill. “Provable Benefits of Actor-Critic Methods for Offline Reinforcement Learning."
> [6]: Jin, Yang, Wang. “Is Pessimism Provably Efficient for Offline RL?”

---

> > ### Comment · Reviewer_9CB2 · 2022-08-03
> > **Thanks for the clarification**
> >
> > I thank the authors for their response. They address my concern about the theoretical contribution of the work. But I still doubt its real-world applicability since real-world problems are rarely "linearly realizable" and might not belong to the benign class of instances analyzed in the paper.
> >
> > I would like to increase my evaluation to 6.

---

### Official Review · Reviewer_oqs5 · 2022-07-11

**Rating:** 7
**Confidence:** 2
**Soundness:** 3 good
**Presentation:** 3 good
**Contribution:** 4 excellent

**Summary:**

This paper proposes a family of offline pessimistic learning algorithms for linear contextual bandit based on $\ell_p (p\geq 1)$ confidence sets (called $\widehat{\pi}_p$). Among them, the algorithm based on $\ell_\infty$ confidence set (called $\widehat{\pi}_\infty$) has the smallest suboptimality gap. Furthermore, it also proves lower bounds for classes of linear contextual bandit problems indexed by $q\in[1, \infty]$ and shows that $\widehat{\pi}_\infty$ is minimax optimal for all classes of problems.

**Questions:**

- It looks peculiar that the requirement for $\Lambda$ in lower bound becomes purely numeric when $p=\infty$. Is there any intuition for this?

**Limitations:**

Yes, the limitations are adequately discussed, especially the regime where the proposed lower bounds are applicable.

**Strengths And Weaknesses:**

### Strengths
The claim that $\widehat{\pi}_\infty$ is adaptively minimax optimal is appreciated and considered to be highly novel. Meanwhile, this paper also provides thorough discussion and comparison between $\widehat{\pi}_\infty$ and previous methods, showing non-trivial advantage of $\widehat{\pi}_\infty$. Furthermore, to support its claim, this paper also proposes a novel lower bound and complexity measure for linear contextual bandit problems.

The writing of the paper is also in a good logic flow.

### Weaknesses

Minor issues:
- It may be better to have a separate section for conclusion to summarize what has been discussed before.
- It may be better to give a sketch or core idea of the hard instance construction for proving the lower bounds.

---

> ### Author Response · Authors · 2022-08-01
> **Thank you for your review.**
>
> We thank the reviewer for their comments and time. To answer the question asked by the reviewer:
>
> **The requirement for $\Lambda$ in the lower bound.** Thanks for the interesting question. We conjecture that it is possible to improve the lower bound to hold for any $\Lambda = \Omega(1)$ for all $p \in [2,\infty)$. Due to technical challenges, we were only able to do so for $p= \infty$. For $p=\infty$, we use a different construction which uses sparsity of the test distribution $\rho$, and it is unclear how to adapt it for $p < \infty$. In Appendix D.1, we do have a more complicated lower bound construction which holds for any $\Lambda = \Omega(1)$ for $p \in [2,\infty)$, but unfortunately it incurs an undesirable stronger requirement on sample complexity $n$ in order for the lower bound to hold, so we did not include it in the main text.
>
> In future revision, we will incorporate some discussion on this in the main text. As you point out, it does look a bit peculiar without any explanation. We can also include more details on the lower bound in the main text to provide intuition.

---

> > ### Comment · Reviewer_oqs5 · 2022-08-04
> > **Response to author feedback**
> >
> > Thank you very much for the clarification and my question is well-addressed. I believe this is a good work and I'll thus keep my score.

---

### Official Review · Reviewer_aDkY · 2022-07-11

**Rating:** 6
**Confidence:** 4
**Soundness:** 3 good
**Presentation:** 4 excellent
**Contribution:** 3 good

**Summary:**

This paper proposed a new confidence set estimation approach for linear contextual bandits based on the pessimistic principal. The optimality gap is derived using the dual norm techniques. The authors are able to show that the l_p confidence set attains the optimal rate when p=infinity. Furthermore, the paper showed that using p=infinity also achieves the adaptive minimax optimality. Experiments on synthetic dataset demonstrate the confidence set derived in this paper is valid and using l_infinity norm indeed outperforms the other candidates.

**Questions:**

(1) How to justify that the paper has significant contributions to the linear contextual bandit community?

**Limitations:**

Yes

**Strengths And Weaknesses:**

Strengths:

(1). This paper provided a suite of methods for deriving valid confidence set in the linear contextual bandit scenario. Furthermore, the paper suggested that using l_p norm with p=infinity gives the best suboptimality guarantee. The theoretical results are interesting, which I believe will facilitate future research in offline linear contextual bandits.

(2). The pessimism principle is well-motivated in the beginning of the paper, and it is very clear why pessimism would work in the offline linear contextual bandit scenario through equation (7).

(3). The related works are thoroughly discussed. Two important prior works, the LCB and BCP, are special instantiations of the general confidence set derived in this paper. Therefore, this paper can be viewed as an extension of prior research to the broadest setting.

Weaknesses:

(1). I am a bit skeptical of the contribution of this paper. In particular, the paper basically unified the prior works on LCB and BCP and provided a generic approach for confidence set estimation. The paper also suggested that when p=infinity, the confidence set estimation and the error rate attains the optimum. However, again as the paper pointed out, LCB is an instantiation of the generic approach when p=infinity, thus is already optimal. Therefore, the contribution of this paper is mostly about a general framework to unify previous methods.

---

> ### Author Response · Authors · 2022-08-01
> **Thank you for your review.**
>
> We thank the reviewer for their comments and time.
>
> **To justify the contribution for the linear contextual bandit community**: prior work on offline linear contextual bandits/RL suggested the BCP method [1], the PACLE method [2], and the PEVI method [3], which in this work we show is suboptimal, and instead our work suggests a different method that we argue strictly dominates these methods. Suggesting a method that strictly dominates the previously published methods seems to us a significant contribution.
>
> The general framework of pessimism via confidence sets is a tool we develop in this paper.  Indeed, once we present this general framework, the relationship to BCP, LCB, PACLE, and PEVI becomes immediate, it seems natural to use an $\ell_\infty$ confidence set also for general linear contextual bandits, and our method just pops out.  We are happy this is all clear after reading the paper and consider this “obviousness-in-hindsight” a good thing.  But it's important to keep in mind the unifying view we present is what enabled this, and this was not obvious, e.g., to researchers who worked on the problem before us and suggested BCP, PACLE, and PEVI.
>
> [1]: Xie, Cheng, Jiang, Mineiro, Agarwal. “Bellman-consistent Pessimism for Offline Reinforcement Learning.”
> [2]: Zanette, Wainwright, Brunskill. “Provable Benefits of Actor-Critic Methods for Offline Reinforcement Learning."
> [3]: Jin, Yang, Wang. “Is Pessimism Provably Efficient for Offline RL?”

---

### Official Review · Reviewer_hhtZ · 2022-07-12

**Rating:** 8
**Confidence:** 2
**Soundness:** 4 excellent
**Presentation:** 4 excellent
**Contribution:** 3 good

**Summary:**

This paper focuses on offline learning for linear contextual bandits and provides a novel family of pessimistic learning rules that generalizes over the Bellman--consistent pessimism and lower confidence bound strategies. The statistical guarantees established here for this new family of learning rules are proven to be minimax optimal, as the authors also show a lower bound. Last is demonstrated the adaptive minimax optimality property of one of the new learning rules - the extension of the lower confidence bound strategy - with empirical experiments corroborating the theoretical findings.

**Questions:**

- Could this work be extended to RL?
- The definition of the policies of interest are only explicitly made in Equation (9), while they should be explicitly defined before Theorem 1.
- Could the confidence set in Equation (8) and Lemma 1 incorporate more instance-dependent terms in order to get tighter results?


**Limitations:**

The theoretical limitations are adequately addressed. The authors state that the potential negative societal impacts of their work is N/A due to its theoretical nature. It might still be valuable to mention what could go wrong if the suggested algorithms were actually deployed.


**Strengths And Weaknesses:**

Strengths:
- Presentation: the problem is well introduced and the main results are clearly presented
- Impact: this paper provides a minimax optimal solution to the problem of offline linear contextual bandits. This new family of learning rules generalizes well-known approaches.
- The paper is technically sound.
- The experiments seem to nicely support the theoretical findings

Weakness: No instance dependent results. It seems like the instance dependent literature for linear is growing (even for RL, see [1,2]) and it would have been to see a result of that form

[1] Zanette, A., Kochenderfer, M. J., and Brunskill, E. Almost horizon-free structure-aware best policy identification with a generative model.
[2] Wagenmaker, A., Simchowitz, M., and Jamieson, K. Beyond no regret: Instance-dependent pac reinforcement learning

---

> ### Author Response · Authors · 2022-08-01
> **Thank you for your review.**
>
> We thank the reviewer for their comments and time. To answer the questions asked by the reviewer:
>
> 1. **Extending to RL**. We believe that the learning rules can also be relevant to RL. In fact, the instantiation of the $\ell_2$ rule (i.e., Bellman-consistent pessimism) has already been applied to RL [1,2]. We can modify the PACLE algorithm from [2] to solve for any $\ell_p$ learning rule by suitably changing the convex program (10) from the paper [2].  However, analyzing the statistical performance in the RL setting is a new challenge, and this could indeed be an interesting direction.
> 2. **Instance-dependent guarantees.** This is an excellent question, and thanks for pointing out two relevant RL papers with instance dependent guarantees. There are multiple desiderata in obtaining instance-dependence guarantees in offline RL. The first is the fine-grained dependence on the behavior policy and the optimal policy, which we have achieved in this work. Compared to the single policy concentrability coefficient, our bound is more fine-grained and problem dependent. The second is the dependence on the reward structure. It is relatively easy to develop instance-dependent bounds when the variance associated with each (s,a) pair is different (though one may need to change the confidence set using appropriate weights). However, the more interesting question is the dependence on the gap structure of rewards - this is the usual meaning of “instance-dependence” in the online setting. This question is of great interest and we leave this one to future work.
> We will add further discussion on instance-dependent guarantees in the revised manuscript to incorporate these points.
>
> [1]: Xie, Cheng, Jiang, Mineiro, Agarwal. “Bellman-consistent Pessimism for Offline Reinforcement Learning.”
> [2]: Zanette, Wainwright, Brunskill. “Provable Benefits of Actor-Critic Methods for Offline Reinforcement Learning."

---

### Meta-Review · Area_Chair_HfL5 · 2022-08-21

**Recommendation:** Accept
**Confidence:** Certain

**Metareview:**

The reviewers are in agreement that this paper provides a minimax optimal solution to the problem of offline linear contextual bandits. This new family of learning rules beat state of the art approaches and provide a unified view on existing approaches, such as Lower Confidence Bound and Bellman-Consistent Pessimism. The theoretical results are backed by reasonable numerical simulations. Accept.

**Award:**

No

---

### Decision · Program_Chairs · 2022-09-14

Accept